# Anxiety and Depression Disorders in Undergraduate Medical Students During the COVID-19 Pandemic: An Integrative Literature Review

**DOI:** 10.3390/ijerph21121620

**Published:** 2024-12-03

**Authors:** Ana Luisa Varrone Sartorao, Carlos Izaias Sartorao-Filho

**Affiliations:** Faculty of Medicine, Educational Foundation of the Municipality of Assis., São Paulo 19815-110, Brazil

**Keywords:** anxiety, depression, medical students, pandemic, COVID-19, GAD-7, PHQ-9, mental health

## Abstract

Introduction: The COVID-19 pandemic has triggered several challenges on the front of mental health. Undergraduate medical students face considerable stress in their academic routines. Thus, there is a need to explore the implications for the mental health of undergraduate medical students during the COVID-19 pandemic. Objective: To review the global literature about anxiety and depressive disorders in undergraduate medical students during the COVID-19 pandemic. Method: We developed an integrative literature review on the occurrence of anxiety and depressive symptoms in undergraduate medical students during the COVID-19 pandemic. We included the manuscripts that used the PHQ-9 and/or GAD-7 questionnaires. We excluded systematic reviews, narrative reviews, integrative reviews, meta-analyses, and qualitative analytical studies. We assessed the results on the occurrence of anxiety and depression and the severity of symptoms in medical students during the COVID-19 pandemic using quantitative studies applying the GAD-7 questionnaire for anxiety or the PHQ-9 for depression. Results: We reviewed 85 selected studies, and the results showed a significant prevalence of moderate and severe symptoms of anxiety and depression, with 28.2% of participants presenting scores of ≥10 on the GAD-7 and 38.9% on the PHQ-9. Statistical analyses using simple and multiple regression tests revealed associations between higher rates of anxiety symptoms among students from developing countries and data collected after the lockdown period in 2020 during the pandemic lockdown. In addition, female students were at risk of depressive disorders. We emphasize as a limitation that the diagnosis of depression and anxiety requires a detailed clinical evaluation, which is not focused on in this actual study. Conclusions: Our findings highlight the need for specific interventions to support the mental health of undergraduate medical students, especially female students from developing countries, during a pandemic crisis.

## 1. Introduction

In December 2019, the first case of a new respiratory disease caused by the SARS-CoV-2 virus was documented [1]. On 11 March 2020, the World Health Organization (WHO)’s executive director officially categorized COVID-19 as a pandemic [2]. In May 2023, the WHO declared an end to the public health emergency of international concern regarding this disease [3,4]. As a result, in response to the global health crisis, the most widely used approach was social isolation, which resulted in the transition of in-person educational activities to an online format [5].

COVID-19 not only causes physical health problems but can also lead to a series of mental disorders [6]. Fear of death and the impacts on physical health, isolation, social distancing, the loss of family members, financial difficulties, misinformation, rumors, and uncertainty about the future are sources of distress. According to the surveys conducted by the WHO, the COVID-19 pandemic triggered a 25% increase in the prevalence of depression and anxiety worldwide [6]. Thus, the effects of the COVID-19 pandemic provide an opportunity to reflect on the state of mental health and highlight the imminent need to implement fundamental preventive measures for collective well-being [6]. Hence, concerns arise regarding the mental health of undergraduate medical students, who represent a population that already suffers from the daily pressures of academic life, which can compromise mental, social, and physical health [7]. Furthermore, there was a higher prevalence of symptoms of anxiety and depression in medical students related to the COVID-19 pandemic [8]. The mental health of medical students presents a vital need to analyze whether depression and anxiety symptoms represent obstacles to their academic careers [9].

Many studies have evaluated scales such as the GAD-7 and PHQ-9 in medical students during the COVID-19 pandemic. The Generalized Anxiety Disorder (GAD-7) scale is a seven-item diagnostic tool that shows probable cases of generalized anxiety disorder and assesses symptom severity. It has been confirmed in remote health surveys, epidemiological studies, and primary care settings [10]. This questionnaire is reliable and has a criterion validity [11]. However, this scale only provides probable diagnoses, which need to be confirmed through further assessment [11]. The Patient Health Questionnaire-9 (PHQ-9) is a nine-item questionnaire that screens for depression in primary care and other medical settings [12]. It is a quick, effective, simple, and reliable tool for screening and assessing the severity of depressive symptoms [13]. However, this questionnaire does not necessarily match the lived experience of depression [14]. Thus, the PHQ-9 is not considered an instrument to confirm a depression diagnosis [15].

Therefore, a literature review that analyzes the rates and severity of depression and anxiety symptoms in undergraduate medical students during the COVID-19 pandemic is essential. Although the emergency phase of this pandemic has already ended, it is essential to analyze the psychological effects on medical students, aiming to provide data that guide the development of strategies for future interventions in similar crises. We hypothesize that there is an increase in the occurrence and severity of symptoms of anxiety and depression in medical students during the COVID-19 pandemic. Hence, we aimed primarily to review the global literature on anxiety and depression disorders with studies that used the PHQ-9 and GAD-7 questionnaires in undergraduate medical students during the COVID-19 pandemic. Moreover, the specific objectives were to analyze the predictive variables for increased symptoms of anxiety and depression in medical educational institutions.

## 2. Methods

We performed an integrative literature review from February to July 2024 at the Faculty of Medicine at FEMA (Educational Foundation of the Municipality of Assis). Regarding the eligibility and search criteria, we included medical literature in English, Portuguese, and Spanish, using the following keywords: (COVID-19) and (Medical Students) and (anxiety) or (depression) or (mental health). We searched the indexed journals’ databases, PubMed and Bvsalud, and selected the manuscripts with data collection from December 2019 to July 2024. We included the manuscripts that used the PHQ-9 and/or GAD-7 questionnaires in their methodology. We excluded systematic reviews, narrative reviews, integrative reviews, meta-analyses, and qualitative analytical studies. We also excluded manuscripts from non-indexed and pre-print journals.

We aimed to identify peer-reviewed literature and relevant studies focusing on symptoms of depressive and anxiety disorders in undergraduate medical students during the COVID-19 pandemic. To ensure a comprehensive search, we searched PubMed and BVSalud databases using keywords, subject headings, and boolean operators. Our terms were carefully selected to focus on the target population, mental health disorders, and the context of the COVID-19 pandemic.

We used the following variables: Studies with data collected in 2020, during the lockdown period, and studies conducted after 2020, post-lockdown period; the continents in which the institutions were located, which were Europe, Asia, North America, Latin America, Oceania, and Africa; the actualized Human Development Index (HDI) of the country; the number of study participants, gender, percentile of women, and the average age of the participants; the categorization of the GAD-7 questionnaire, organized into scores of 0–4, 5–9, 10–14, 15–21, and the categorization of the GAD-7 was <10 or ≥10; the categorization of the PHQ-9 questionnaire, organized into scores of 0–4, 5–9, 10–14; 15–19 > 19 and the categorization of the PHQ-9 was <10 or ≥10. The primary outcome was the prevalence of moderate or higher symptoms of anxiety and depression in medical students during the COVID-19 pandemic. The secondary outcome was a search for predictive variables on the severity of symptoms in studies in which the GAD-7 and PHQ-9 scores were ≥10.

We adjusted the multiple linear regression models with a normal response to explain a GAD-7 percentage that was greater than or equal to 10 points and a PHQ-9 percentage greater than or equal to 10 points, including, in the deterministic component, only the variables that presented a *p* < 0.20 in the bivariate investigation. The GAD-7 and PHQ-9 indexes equal to or greater than ten are critical for denoting moderate- and high-severity symptoms [11,16]. The quality of the adjustment of the multiple regression models was analyzed by investigating the behavior of the residuals with the Shapiro–Wilk normality test, scatter plot between residuals and predicted values of the models to investigate homoscedasticity, and Cook’s distance measure to investigate the influence of atypical points on the estimates of the model parameters. The final models considered the associations statistically significant if *p* < 0.05. All analyses were performed with the SPSS 21 software by IBM trademark.

We collected studies from the literature using secondary data sources. We were concerned about bias risks in the analysis and the interpretation of research data and indirect risks to the physical, mental, spiritual, and social dimensions associated with human beings in any research. Regarding the benefits, we considered the manuscript vital for promoting a positive impact on medical practice in mental health.

## 3. Results

We identified 1768 studies in the PubMed database and 60 articles in Bvsalud. Applying the eligibility criteria, we selected 88 articles from the literature. Of these, 43 contained at least one of the questionnaires (GAD-7 or PHQ-9), while in 42 articles, both questionnaires were used. Figure 1 represents the flowchart of the selection process.

Table 1 contains the list of selected studies. Three manuscripts were excluded because they were published in non-indexed databases. Thus, this study was based on data from 85 manuscripts in the literature.

Table 2 shows the manuscripts’ sample characteristics. Of the selected manuscripts, 28 collected information after 2020. Regarding the institution continent area, 49 papers were from Asian institutions. For the HDI (Human Development Index), 35 were from countries with a very high HDI, 31 were from countries with a high HDI, and 22 were from countries with a medium or low HDI.

Table 3 provides the variable range list and the interquartis. Concerning the HDI variable, the median of the countries was 0.79 (0.70–0.88). The median age of the participants was 22.0 years (20.0–23.0). The percentile of women who answered the questionnaires was 63.0% (52.3–68.7). Regarding the GAD-7 questionnaire, we observed a median of 28.2% (18.3–39.4) with a score of ≥10. Regarding the PHQ-9 questionnaire, the median score of ≥10 was 38.9% (26.8–47.2).

Table 4 presents the bivariate associations by simple linear regression to explain the percentage of GAD-7 with a score of ≥10 points (*p* < 0.20). We observed significant results (*p* < 0.20) regarding the following variables: data collected in 2020, Latin America, Oceania, and Asia (reference: Europe), with a low, medium, or high HDI (reference: very high HDI).

Table 5 shows the data obtained after multiple linear regression to explain the percentage of GAD-7 scores that were ≥10 (*p* < 0.05). After statistical analysis, we observed that studies with data collected in 2020—during the lockdown in most countries worldwide—had a GAD-7 response percentage of ≥10, which was, on average, 14% lower compared to the data collected after 2020 (β: −14.02; 95% CI −21.63 to −6.40; *p* < 0.001). Countries with a medium or low HDI had a GAD-7 response percentage of ≥10, twelve percent higher than countries with a high or very high HDI (β: 12.61; 95%CI 2.93 to 22.29; *p* < 0.011).

Regarding the studies using the PHQ-9, Table 6 shows the bivariate associations by simple linear regression to explain the percentage of PHQ-9 scores ≥10. As a result, the percentage of women was the only association presenting a *p*-value under 0.20 on the bivariate analysis (β: 0.36; 95%CI: −0.04 to 0.75; *p* < 0.077). For the PHQ-9 analysis, the multiple linear regression was not carried out because the analysis was inappropriate due to only one significant variable detected by the bivariate analysis. Thus, the multivariate analysis would require multiple predictors to provide value.

## 4. Discussion

We found evidence that studies performed in 2020 during the lockdown period showed students with an average reduction of 14% in their responses to the GAD-7 anxiety symptoms score of ≥10 (moderate and severe symptoms) compared to studies after 2020. The lockdown period and the beginning of the pandemic, as well as a period of uncertainty, may explain this finding [104,105].

Furthermore, we observed that students from developing nations had a significant average increase of 12% in their responses to moderate or severe symptoms of anxiety than those from countries with a high or very high HDI. Moreover, regarding the higher percentage of female students, a significant association was found concerning the PHQ-9 depression symptoms score of ≥10. It is known that females have a predisposition to be more affected by stressors related to the COVID-19 pandemic, such as social isolation, academic disruptions, and health concerns [106,107]. Moreover, it is essential to highlight that developing countries are also associated with a high burden of mental health disorders [108], with some studies suggesting that lower-income countries have a reduced capacity to provide access to depression treatment [109,110].

The COVID-19 pandemic has affected medical students’ mental health. A systematic review found exacerbated feelings of stress, depression, and anxiety among medical students worldwide [111]. Another study reported that 23% of medical students experienced depression, and 11% experienced anxiety during the pandemic [112]. A cross-sectional survey revealed that a staggering 79.6% of medical students experienced heightened anxiety, and 65.1% reported increased depression during the pandemic [54]. Similarly, a Serbian study found that a significant majority of medical students, 64.5%, reported severe depressive symptoms, 66.8% reported severe anxiety, and 66.7% reported severe stress [113].

COVID-19-related mental health issues more significantly affect female medical students, including increased symptoms of depression and anxiety. A study conducted in the United States observed that the female group was significantly more likely to report anxiety risk symptoms among medical students [114].

Another meta-analysis identified that being female, being a junior, being a preclinical student, and having economic troubles were significant risk factors for mental health impairment during the pandemic [115]. 

In Mexico, factors associated with depression included the female sex, younger age (18–20 years), perceived academic performance, and economic hardship [116].

A meta-analysis published by Jia et al. in 2022 demonstrated the pooled prevalence of depression in 37.9% of medical students (95%CI: 30.7–45.4%) and pooled anxiety prevalence of 33.7% (95%CI: 26.8–41.1%). In addition, their results varied by gender, country, and continent [117].

Another study, in 2024, from Lin et al. [118],reported a pooled prevalence for anxiety of 45% (95%CI: 40–49%) and depression of 48% (95%CI: 43–52%). For moderate and severe anxiety, 28% (95%CI: 24–32%), and for moderate and severe depression, 30% (95%CI: 26–35%). After the meta-regression, medical students in Asia had a lower prevalence of anxiety and depression than those from other regions [118].

Limitations: The review analyzed global data based on observational studies using the questionnaires GAD-7 and PHQ-9, which are recognized as good screening instruments for anxiety and depression symptoms. These questionnaires are widely used in mental health research due to their reliability and validity. We only elected studies using GAD-7 for anxiety and PHQ-9 for depressive disorders because the diversity of tools used across studies can lead to difficulties in comparing and synthesizing results. In addition, the diagnosis of depression and anxiety is not based solely on the application of the questionnaires and requires a detailed clinical evaluation. Other limitations were the heterogeneity of the population, variability in the study design, publication bias, local or context-specific stressors, and cultural and subjective influence on the responses. Moreover, the co-occurrence of depression or anxiety may lead to confounding results in the observational studies. Lastly, dichotomous variables often reduce nuanced information into two categories, which can oversimplify complex relationships in the statistical regression model.

The results cannot be generalized due to the consideration of limitations inherent to observational studies, such as difficulty in controlling variables, potential confounders, temporal ambiguity, and the location where the studies were conducted, in addition to selection and information biases. Overall, our findings underscore the heightened mental health difficulties faced by medical students during the pandemic, highlighting the essential need for targeted support and preventable interventions.

## 5. Conclusions

A comprehensive review of global literature demonstrated a high occurrence of moderate and severe symptoms of depression and anxiety among the undergraduate medical student population. We observed a higher occurrence of anxiety symptoms among medical students assessed after the lockdown period and from studies assessed in developing countries. We also described a higher occurrence of depressive symptoms in the female population worldwide. These findings highlight the urgency of developing targeted intervention strategies to mitigate these symptoms in populations that demonstrate high susceptibility to mental disorders during pandemic periods.

## Figures and Tables

**Figure 1 ijerph-21-01620-f001:**
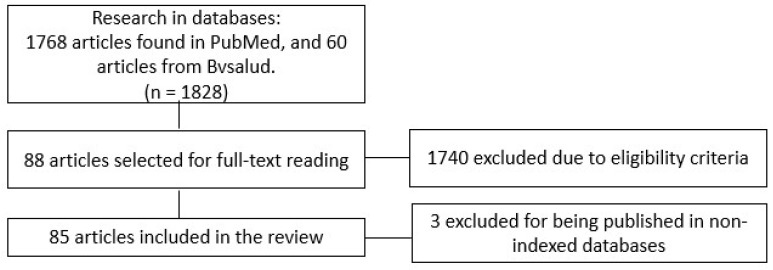
Flowchart of the selection process of the studies included in the review.

**Table 1 ijerph-21-01620-t001:** List of the eligible studies and the percentiles of GAD-7 and/or PHQ-9 scores ≥10.

	Author	Country	N	GAD-7 ≥ 10 (%)	PHQ-9 ≥ 10 (%)
1	Zheng [17]	China	468	11.30	20.70
2	Coico-Lama [18]	Peru	431	29.50	28.50
3	Bhongade [19]	Emirates	107	25.30	
4	Din [20]	Pakistan	444	46.17	64.41
5	Reddy [21]	India	164	20.00	
6	Ortega-Moreno [22]	Mexico	384	24.50	43.00
7	Shahzad [23]	Pakistan	585	41.00	
8	Iqbal [24]	India	261	51.70	58.70
9	Gomez-Duran [25]	Spain	175	34.70	26.60
10	Wiguna [26]	Indonesia	1023		77.40
11	Tanuseriawan [27]	Indonesia	635		63.40
12	Purnomo [28]	Indonesia	161		8.70
13	Yuryeva [29]	Ukraine	154	27.90	44.80
14	Arshad I [30]	India	261	65.50	67.80
15	Lakshmi [31]	India	200	83.00	
16	Ernst J [32]	Swiss	574	22.60	
17	Cao W [33]	China	7143	3.60	27.20
18	Chistophers B [34]	USA	1139	20.00	
19	Sartorao [8,35]	Brazil	340	^a^	^a^
20	Lin S [36]	USA	154		24.00
21	Huarccaya Victoria [37]	Colombia	1238		34.00
22	Pinsai [38]	Tailandia	37	51.35	
23	Verma [39]	India	267	28.50	
24	Alkwai [40]	Saudi Arabia	55	17.00	26.42
25	Bartra [41]	Peru	57	22.80	
26	Guralwar [42]	India	604	54.14	
27	Almarri [43]	Saudi Arabia	7116	40.50	
28	Kamran [44]	Pakistan	324	44.50	
29	Porwal [45]	Saudi Arabia	22	13.60	40.90
30	Primatanti [46]	Indonesia	7949	13.90	
31	AbuDujain [47]	Saudi Arabia	345	13.90	
32	Imran [48]	Pakistan	1100	40.40	48.10
33	Rafsanjanipoor [49]	Iran	83	24.20	
34	Srivastava [50]	India	97	24.74	48.10
35	Pedraz-Petrozzi [51]	Colombia	125	12.80	34.40
36	Vajpeyi [52]	Emirates	798	39.10	
37	Alshehri [53]	Saudi Arabia	182	30.80	
38	Paz D [54]	USA	152	36.70	40.90
39	Schindler [55]	Germany	63		44.00
40	Lu [56]	China	519		41.50
41	Chakeyanunn [57]	Thailand	437		27.00
42	Huarcaya victoria [37]	Colombia	1238	19.00	
43	Camelier-Mascarenhas [58]	Brazil	310	33.50	42.60
44	Dziedzic [59]	Brazil	162	29.60	34.00
45	Eleftheriou [60]	Greece	559	67.60	43.70
46	Cheng [61]	China	947	37.80	39.30
47	Santander [62]	Peru	370	38.38	
48	Çimen [63]	Turkey	2778	44.50	46.21
49	VIillalon López [64]	Chile	359	41.50	60.10
50	Villagomes-Lopez [65]	Ecuador	1528	30.30	
51	Harries [66]	USA	741	25.60	
52	Liu [67]	China	29,663	46.00	37.80
53	Pattanaseri [68]	Thailand	224	^a^	^a^
54	Teh [69]	Malaysia	371	37.00	35.70
55	Adhikari [70]	Nepal	223	^a^	^a^
56	Chalise [71]	Nepal	315	12.90	
57	Romic [72]	Croatia	280	32.50	52.20
58	Nguyen [73]	Vietnan	747	7.90	20.63
59	Biswas [74]	Bangladesh	425		31.80
60	Song [75]	China	666	17.80	15.20
61	Guo [76]	USA	929	31.10	48.80
62	Essangri [77]	Morocco	549	25.70	45.70
63	Saali [78]	USA	108	32.40	
64	Nishimura [79]	Japan	473	7.20	74.70
65	Sserunkuuma [80]	Uganda	269		24.10
66	Batais [81]	Saudi Arabia	332	13.70	15.90
67	Crisol-deza [82]	Peru	1238	19.00	34.00
68	Tsiouris [83]	Germany	1438		34.00
69	Sudi [84]	Malaysia	196		38.90
70	Wercelens [85]	Brazil	150		40.70
71	Yin [86]	China	5982	4.20	9.90
72	Chwa [87]	USA	87	27.40	
73	Pandey [88]	India	83	9.80	24.70
74	Elhadi [89]	Libya	2430	27.00	
75	Xiao [90]	China	933	4.60	7.30
76	Essadek [91]	France	668		42.80
77	Liu [92]	China	217	7.40	
78	Chootong [93]	Thailand	325	12.90	7.60
79	Saeed [94]	Pakistan	234	62.40	64.10
80	Huang [95]	China	1021	10.98	38.17
81	Wang [96]	Korea	454	18.50	11.10
82	Halperin [97]	USA	1428	30.60	31.00
83	Bilgi [98]	Turkey	178	37.10	20.10
84	Alsairafi [99]	Kuwait	298	85.20	93.00
85	Allah [100]	Saudi Arabia	1591	19.20	
86	Khidri [101]	Pakistan	864		40.80
87	Shreevastava [102]	India	1208	40.30	
88	Afzal [103]	Pakistan	433		40.65

^a^ The manuscripts referenced as 19, 53, and 55 were excluded due to non-indexed publication.

**Table 2 ijerph-21-01620-t002:** Manuscripts’ description variables.

Variable	*N*	*%*
**Data collection time**		
After 2020	28	31.8
In 2020	60	68.2
**Continent**		
Europe	9	10.2
North America	8	9.1
Asia	49	55.7
Oceania	5	5.7
Latin America	14	15.9
Africa	3	3.4
**Human Development Index (HDI)**		
Very high	35	39.8
High	31	35.2
Medium or low	22	25.0

**Table 3 ijerph-21-01620-t003:** Variable range list and the interquartile ranges.

Variable	Median	Q1	Q3
Human Development Index	0.79	0.70	0.88
Number of participants	377.5	185.5	912.8
Male	160.0	89.0	322.0
Female	240.0	113.0	597.0
Percentual of women	63.0	52.3	68.7
Age	22.0	20.0	23.0
GAD-7 score 0–4	25.3	0.0	39.2
GAD-7 score 5–9	37.8	30.4	67.2
GAD-7 score 10–14	19.9	12.8	27.5
GAD-7 score 15–21	3.4	0.0	13.9
GAD-7 score ≥ 10	28.2	18.3	39.4
PHQ-9 score 0–4	0.0	0.0	28.4
PHQ-9 score < 10	40.0	27.0	60.9
PHQ-9 score 10–14	23.0	19.0	36.8
PHQ-9 score 15–19	4.9	0.0	13.9
PHQ-9 score > 19	0.0	0.0	6.2
PHQ-9 score ≥ 10	38.9	26.8	47.2

Q1: first interquartile range. Q3: third interquartile range.

**Table 4 ijerph-21-01620-t004:** Bivariate associations by simple linear regression to explain the percentage of GAD-7 scores ≥10.

Variable	*β*	95%CI		*p*
Data Collection in 2020 (Ref: After 2020)	−15.16	−23.15	−7.17	0.000
Africa	−13.48	−40.54	13.58	0.329
Latin America	−13.22	−29.79	3.35	0.118
Oceania	−25.93	−61.73	9.87	0.156
Asia	−9.51	−23.98	4.95	0.198
North America	−10.72	−29.16	7.72	0.255
Continent (ref: Europe)	0			
Human Development Index	−34.12	−68.59	0.35	0.052
Medium or low	9.23	−0.12	18.59	0.053
High	−10.72	−19.10	−2.35	0.012
Human Development Index (ref: very high)	0			
Number of participants	0.00	0.00	0.00	0.914
Number of women	0.00	0.00	0.00	0.963
Percentage of women	0.08	−0.30	0.46	0.684
Average age	0.67	−3.82	5.15	0.771

***β***: beta coefficient; 95%CI: 95% confidence interval; *p* < 0.20.

**Table 5 ijerph-21-01620-t005:** Multiple linear regression to explain the percentage of GAD-7 scores ≥10.

Variable	*β*	95%CI		*p*
Data collection in 2020 (ref: after 2020)	−14.02	−21.63	−6.40	*0.000*
Africa	−6.26	−30.28	17.76	*0.610*
Latin America	2.63	−13.92	19.18	*0.755*
Oceania	−22.24	−53.37	8.90	*0.162*
Asia	−7.25	−20.49	5.99	*0.283*
North America	−5.38	−20.71	9.96	*0.492*
Continent (ref: Europe)	0			
Medium or low	12.61	2.93	22.29	*0.011*
High	−8.37	−18.37	1.63	*0.101*
Human Development Index (ref: very high)	0			

*p* < 0.05; homoscedasticity; dCook < 1 = 100%; ***β***: beta coefficient; 95%CI: 95% confidence interval.

**Table 6 ijerph-21-01620-t006:** Bivariate associations by simple linear regression to explain the percentage of PHQ-9 scores ≥10.

Variable	*β*	95%CI		*p*
**Data Collection in 2020 (Ref: After 2020)**	1.42	−8.71	11.54	0.784
Africa	−8.59	−35.82	18.64	0.536
Latin America	−4.46	−21.19	12.28	0.602
Oceania	3.61	−17.48	24.70	0.737
Asia	−6.21	−19.92	7.49	0.374
North America	−7.31	−28.40	13.78	0.497
Continent (ref: Europe)	0			
Human Development Index	−22.58	−62.59	17.42	0.269
Medium or low	6.67	−5.34	18.68	0.276
High	−6.14	−16.19	3.92	0.232
Human Development Index (ref: very high)	0			
Number of participants	0.00	0.00	0.00	0.625
Number of women	0.00	0.00	0.00	0.665
Percentage of women	0.36	−0.04	0.75	0.077
Average age	4.78	−5.13	14.70	0.344

***β***: beta coefficient; 95%CI: 95% confidence interval; *p* < 0.20.

## Data Availability

Not applicable.

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
