# Peer review of "Anxiety and Depression Disorders in Undergraduate Medical Students During the COVID-19 Pandemic: An Integrative Literature Review"

_ijerph, 2024, doi:10.3390/ijerph21121620_

Round 1
Reviewer 1 Report
Comments and Suggestions for Authors
1. In line 9 in the abstract, remove "the" before the word "mental health".
2. In line 47, avoid referring to your institution and use appropriate language to maintain objectivity. Since your study has a global scale, consider adding more references from different countries worldwide showing the impact of the COVID-19 pandemic on undergraduate medical students.
3. It is not clear in the search strategy which is your target population. Please clarify.
4. In line 124, the total number of studies is written as a decimal. Please correct the number.
5. Table 1 on study characteristics lacks important information such as sample size, target population, setting, mental health problems assessed, year in which the study took place, etc. I would recommend expanding the information listed in that table. Table 2 should be merged with table 1 as specific columns listing the number of variables collected in each study. You can also stratify studies by low, middle, and high-income countries in which they took place.
6. Is there a reason why multiple linear regression was carried out for GAD-7 but not PHQ-9?
7. The discussion lacks rigor. It lists the findings but does not support sufficient evidence from the global literature to support the results. Additionally, it does not provide any recommendations on how to address such a global public health problem in medical students.
Comments on the Quality of English Language
Grammatical errors are apparent across the manuscript.
Author Response
Reviewer 1
- In line 9 in the abstract, remove "the" before the word "mental health".
We are pleased to respond to the suggested amendments. We performed a grammatical revision in the manuscript and we removed “the” from the sentence. Our apologies for the mistake.
- In line 47, avoid referring to your institution and use appropriate language to maintain objectivity. Since your study has a global scale, consider adding more references from different countries worldwide showing the impact of the COVID-19 pandemic on undergraduate medical students.
Thank you for pointing this out. We reviewed the sentence in line 47.
- It is not clear in the search strategy which is your target population. Please clarify.
Thank you for the suggestion. In the method section, we added a new sentence to clarify the search strategy.
- In line 124, the total number of studies is written as a decimal. Please correct the number.
Our apologies for the digitation error. Thanks for pointing this out.
- Table 1 on study characteristics lacks important information such as sample size, target population, setting, mental health problems assessed, year in which the study took place, etc. I would recommend expanding the information listed in that table. Table 2 should be merged with table 1 as specific columns listing the number of variables collected in each study. You can also stratify studies by low, middle, and high-income countries in which they took place.
Table 1 is very extensive. Merging the tables would distract the reader's attention, as Table 1 already contains a substantial amount of data. Therefore, we chose to leave the two tables separate to reduce this aforementioned risk and maintain the organization of the information.
Thank you for this suggestion. It would have been interesting to explore this aspect. However, in the case of our study, we gently ask you not to merge the tables.
- Is there a reason why multiple linear regression was carried out for GAD-7 but not PHQ-9?
Multiple linear regression is used when there are two or more independent (predictor) variables being analyzed to predict the value of a dependent (outcome) variable. The purpose of using multiple linear regression is to understand the relationship between several predictors and the outcome, accounting for potential confounding or interactions between variables. For GAD-7, we had two or more predictor variables using the bivariate analysis. Thus, the multiple linear regression was mandatory.
For the PHQ-9, however, when only one variable was detected as significant in the bivariate analysis, the following reasons explain why multiple linear regression was not carried out:
- Multiple linear regression requires at least two or more independent variables to be meaningful. If only one variable is significant in the bivariate analysis, there is no need for a multiple regression model. Instead, a simple linear regression can be used, which models the relationship between the single predictor and the outcome.
- Running a multiple regression with just one predictor is redundant because it would yield the same results as a simple linear regression. In other words, the output would not provide any additional insight beyond what was already obtained from the bivariate analysis (e.g., correlation or simple regression).
- The strength of multiple linear regression comes from its ability to control for and assess the effects of multiple variables simultaneously. When only one variable is involved, this control isn't necessary, and no additional predictors are available to improve the model fit, explain more variance, or account for confounding variables.
- If additional non-significant variables are forcefully included in the regression model, this may lead to overfitting, where the model becomes overly complex and tries to capture noise in the data rather than the true underlying relationship.
In summary, multiple linear regression is inappropriate when only one variable is significant in a bivariate analysis because the analysis requires multiple predictors to provide value. In such a case, a simple linear regression is the appropriate method.
- The discussion lacks rigor. It lists the findings but does not support sufficient evidence from the global literature to support the results. Additionally, it does not provide any recommendations on how to address such a global public health problem in medical students.
We agree with this and have incorporated your suggestion throughout the manuscript. We reformulate the discussion and conclusion to adequate and give more support for the results.
We look forward to hearing from you in due time regarding our submission and to respond to any further questions and comments you may have.
Sincerely,
Carlos Izaias Sartorao Filho, MD, PhD
Corresponding author
carlos.sartorao@unesp.br
Reviewer 2 Report
Comments and Suggestions for Authors
-
· The abstract should be written more clearly and concisely, providing specific details about the objectives, methodology, and key findings. Currently, the information is vague and lacks precision, which may hinder the reader’s comprehension.
· The manuscript must elaborate on the differentiation between Anxiety and Depression Disorders in the context of psychiatric illnesses. It is essential to detail how these conditions manifest differently and the implications for treatment and diagnosis in psychiatric patients.
· A section on the limitations of the study should be added, addressing potential biases, methodological constraints, and any factors that may affect the generalizability of the results. Furthermore, a subsection outlining future research directions is necessary to highlight areas that require further investigation and improvement based on the current study's limitations.
· The methodology needs enhancement by incorporating additional diagnostic tests to ensure the robustness and reliability of the findings. Recommended tests include the Beck Depression Inventory-II (BDI-II) for assessing depression severity and the Hamilton Anxiety Rating Scale (HAM-A) for evaluating generalized anxiety. These tests will provide a more comprehensive assessment and strengthen the methodological rigor of the study.
· The sample size determination should be clarified. The results of a G*Power analysis must be included in the manuscript to justify the number of participants and to confirm that the study has sufficient power to detect statistically significant differences or relationships.
· The discussion section would benefit from a comparison of the current study’s findings with those from previous research in the field. A table summarizing relevant studies from the literature, along with their key findings, would provide context and allow for a clearer understanding of how this study contributes to existing knowledge.
· A table of abbreviations should be added to the manuscript, listing all abbreviations used throughout the paper. This will improve the clarity and accessibility of the text for readers who may not be familiar with the terminology.
· The methodology section is not sufficiently rigorous, relying solely on two self-report questionnaires (GAD-7 and PHQ-9). This approach is inadequate for diagnosing anxiety and depression with clinical certainty. The study should be criticized for its lack of clinical interviews or additional psychometric instruments that would validate the findings more reliably. Introducing further diagnostic tools would enhance the overall validity of the results.
· Page 1, Line 9: "on the mental health" → "to mental health."
· Page 1, Line 21: "in developing countries" → "among students from developing countries."
· Page 1, Line 23: "critically in the female students" → "especially in female students."
· Page 3, Line 103: "equal or more than" → "equal to or greater than" and "for denote" → "for denoting."
· Page 2, Line 58: "depression symptoms" → "depressive symptoms."
Author Response
Thank you for allowing us to submit a revised draft of the manuscript titled "Anxiety and Depression Disorders in Undergraduate Medical Students during the COVID-19 Pandemic: An Integrative Literature Review."
We appreciate the time and effort you and the reviewers have dedicated to providing valuable feedback on our manuscript.
We are grateful to the reviewers for their insightful comments on our paper.
We have incorporated changes to reflect most of the suggestions provided by the reviewers.
We have highlighted the changes within the manuscript.
Here is a point-by-point response to the reviewers' comments and concerns.
REVIEWER 2
1) The abstract should be written more clearly and concisely, providing specific details about the objectives, methodology, and key findings. Currently, the information is vague and lacks precision, which may hinder the reader’s comprehension.
We agree with this and have incorporated your suggestion throughout the abstract.
We reviewed and used the PRISMA statement for a better comprehension.
2) The manuscript must elaborate on the differentiation between Anxiety and Depression Disorders in the context of psychiatric illnesses. It is essential to detail how these conditions manifest differently and the implications for treatment and diagnosis in psychiatric patients.
We reviewed the English language and the grammatical quality was improved to provide a better understand on these topics. Our apologies for these inconveniences.
3)· A section on the limitations of the study should be added, addressing potential biases, methodological constraints, and any factors that may affect the generalizability of the results. Furthermore, a subsection outlining future research directions is necessary to highlight areas that require further investigation and improvement based on the current study's limitations.
We are grateful for the instructions. The limitation section was reformulated on the discussion addressing the potential biases and methodological constraints information.
4)· The methodology needs enhancement by incorporating additional diagnostic tests to ensure the robustness and reliability of the findings. Recommended tests include the Beck Depression Inventory-II (BDI-II) for assessing depression severity and the Hamilton Anxiety Rating Scale (HAM-A) for evaluating generalized anxiety. These tests will provide a more comprehensive assessment and strengthen the methodological rigor of the study.
We emphasized and added on method, discussion, and limitation that we considered only eligible those manuscripts that used the GAD-7 and / or the PHQ-9 questionnaires for this review.
5) The sample size determination should be clarified. The results of a G*Power analysis must be included in the manuscript to justify the number of participants and to confirm that the study has sufficient power to detect statistically significant differences or relationships.
We considered that calculating G*Power for an integrative review with over 80 manuscripts was unnecessary because integrative reviews' goals, methods, and nature differ fundamentally from those of quantitative studies that require power analysis for hypothesis testing. Furthermore, the focus is on synthesizing a broad range of studies to offer insights and a comprehensive understanding of the topic.
6) The discussion section would benefit from a comparison of the current study’s findings with those from previous research in the field. A table summarizing relevant studies from the literature, along with their key findings, would provide context and allow for a clearer understanding of how this study contributes to existing knowledge.
We are honored for the commentary. We improved the discussion section adding relevant current studies about the theme.
7) A table of abbreviations should be added to the manuscript, listing all abbreviations used throughout the paper. This will improve the clarity and accessibility of the text for readers who may not be familiar with the terminology.
Thanks for the advice; we added a list of abbreviations at the end of the manuscript.
8) The methodology section is not sufficiently rigorous, relying solely on two self-report questionnaires (GAD-7 and PHQ-9). This approach is inadequate for diagnosing anxiety and depression with clinical certainty. The study should be criticized for its lack of clinical interviews or additional psychometric instruments that would validate the findings more reliably. Introducing further diagnostic tools would enhance the overall validity of the results.
Thank you for your feedback. After careful consideration, we believe that retaining the exclusive focus on the GAD-7 (Generalized Anxiety Disorder-7) and PHQ-9 (Patient Health Questionnaire-9) in our integrative review is justified for the following reasons: Both GAD-7 and PHQ-9 are widely recognized and validated instruments used in clinical and research settings to evaluate anxiety and depressive disorders. Their psychometric properties, including high reliability and validity, have been confirmed across various populations and settings. They are consistent, reliable, and effective for detecting and assessing the severity of these conditions. Including additional instruments may unnecessarily complicate the analysis, especially given the robustness of these tools in meeting the review's objectives.
Moreover, expanding the scope to include other instruments would dilute the review's focus. Hence, our review, which encompasses more than 80 studies, provides robust evidence regarding using these two instruments. The breadth and depth of the existing research on these questionnaires are sufficient to provide meaningful insights into their performance and utility. Adding other instruments might require extending the review without providing additional value or significantly altering the conclusions.
Comments on the Quality of English Language
- Page 1, Line 9: "on the mental health" → "to mental health."
- Page 1, Line 21: "in developing countries" → "among students from developing countries."
- Page 1, Line 23: "critically in the female students" → "especially in female students."
- Page 3, Line 103: "equal or more than" → "equal to or greater than" and "for denote" → "for denoting."
- Page 2, Line 58: "depression symptoms" → "depressive symptoms."
Again, thanks for the comments. Regarding the suggestion to improve the English quality, we acknowledge that clear and precise language is essential for conveying the research effectively. While we have made efforts to ensure the clarity and accuracy of the text, we understand that there may still be areas that require refinement. Thus, we carefully revised the manuscript to enhance the quality of the language and ensure the writing meets the high standards expected by the journal.
We look forward to hearing from you in due time regarding our submission and to respond to any further questions and comments you may have.
Sincerely,
Carlos Izaias Sartorao Filho, MD, PhD
Corresponding author
carlos.sartorao@unesp.br
Reviewer 3 Report
Comments and Suggestions for Authors
This paper aims to review the global literature on anxiety and depression disorders with studies that used the PHQ-9 and GAD-7 questionnaires in undergraduate medical students during the COVID-19 pandemic. Although there is a considerable amount of research on the subject of the study (the effects of COVID-19 on medical students), I think that the current study is worth examining because it is based on a review study.
It would be beneficial to enhance the introduction section of the manuscript with additional literature and to articulate the research questions with greater clarity at the end of the introduction. This will enable readers to gain a clearer understanding of the research questions addressed in the remainder of the study.
It is deemed that the methodology section of the research is adequately elucidated. Furthermore, the tables utilized in the study and the accompanying explanations are deemed sufficient.
It is beneficial that the selection process criteria of the studies included in the study are explained in detail. However, the conclusion and discussion section is the weakest part of the study. Despite the extensive research on the effects of the COIVD-19 pandemic, the discussion section is insufficient. This section requires further development.
In some sections of the study, there are instances where improvements can be made in terms of academic writing and expression styles. For instance, in two separate instances within the study, the authors have written: Furthermore, according to a research developed at our Brazilian medical institution in 2020, there was a higher prevalence of symptoms of anxiety and depression in medical students related to the COVID-19 pandemic page 2, first paragaraph. And on page 8 third paragraph "A study conducted at our Brazilian institution in 2020 applied the GAD-7 and PHQ- 196 9 questionnaires to medical students during the beginning of the COVID-19 pandemic." The expression ‘our brazilian’ in the mentioned sections is not an appropriate academic writing style for an international journal. It would be more correct to rephrase it as In Brazil.
At the end of the discussion section of the study, the researchers articulated the constraints inherent to the study. However, it would be more precise to present this information in a standalone paragraph with the title "Limitations."
Author Response
Dear Editor, Sr. Marko Drasler
Thank you for allowing us to submit a revised draft of the manuscript titled "Anxiety and Depression Disorders in Undergraduate Medical Students during the COVID-19 Pandemic: An Integrative Literature Review."
We appreciate the time and effort you and the reviewers have dedicated to providing valuable feedback on our manuscript.
We are grateful to the reviewers for their insightful comments on our paper.
We have incorporated changes to reflect most of the suggestions provided by the reviewers.
We have highlighted the changes within the manuscript.
Here is a point-by-point response to the reviewers' comments and concerns.
Reviewer 3
This paper aims to review the global literature on anxiety and depression disorders with studies that used the PHQ-9 and GAD-7 questionnaires in undergraduate medical students during the COVID-19 pandemic. Although there is a considerable amount of research on the subject of the study (the effects of COVID-19 on medical students), I think that the current study is worth examining because it is based on a review study.
It would be beneficial to enhance the introduction section of the manuscript with additional literature and to articulate the research questions with greater clarity at the end of the introduction. This will enable readers to gain a clearer understanding of the research questions addressed in the remainder of the study.
We reviewed and enhanced the introduction section. Thank you for pointing this out.
It is deemed that the methodology section of the research is adequately elucidated. Furthermore, the tables utilized in the study and the accompanying explanations are deemed sufficient.
Our gratitude for your comments.
It is beneficial that the selection process criteria of the studies included in the study are explained in detail. However, the conclusion and discussion section is the weakest part of the study. Despite the extensive research on the effects of the COIVD-19 pandemic, the discussion section is insufficient. This section requires further development.
We reviewed and added critical and substantial improvements in the discussion and conclusion section.
In some sections of the study, there are instances where improvements can be made in terms of academic writing and expression styles. For instance, in two separate instances within the study, the authors have written: Furthermore, according to a research developed at our Brazilian medical institution in 2020, there was a higher prevalence of symptoms of anxiety and depression in medical students related to the COVID-19 pandemic page 2, first paragaraph. And on page 8 third paragraph "A study conducted at our Brazilian institution in 2020 applied the GAD-7 and PHQ- 196 9 questionnaires to medical students during the beginning of the COVID-19 pandemic." The expression ‘our brazilian’ in the mentioned sections is not an appropriate academic writing style for an international journal. It would be more correct to rephrase it as In Brazil.
Our gratitude for your suggestions. We reviewed and improved the English language and the suggested sentences above.
At the end of the discussion section of the study, the researchers articulated the constraints inherent to the study. However, it would be more precise to present this information in a standalone paragraph with the title "Limitations."
We added a better explanation and dedicated an exclusive paragraph discussing the limitations.
We look forward to hearing from you in due time regarding our submission and to respond to any further questions and comments you may have.
Sincerely,
Carlos Izaias Sartorao Filho, MD, PhD
Corresponding author
carlos.sartorao@unesp.br
Round 2
Reviewer 1 Report
Comments and Suggestions for Authors
The discussion is still in need of major restructuring. It currently is not coherent and lacks organization.
Author Response
"The discussion is still in need of major restructuring. It currently is not coherent and lacks organization."
Dear reviewer, we are delighted to provide the amendments for your comments in round 2:
We restructured the discussion to clarify the sentences and give a better-structured discussion, following the PRISMA statement as below:
23a Provide a general interpretation of the results in the context of other evidence.
23b Discuss any limitations of the evidence included in the review. 23c
Discuss any limitations of the review processes used.
23d Discuss the implications of the results for practice, policy, and future research.
We expect that the amendments will be approved now. Our gratitude for the revision.
Reviewer 2 Report
Comments and Suggestions for Authors
The article can be accepted in its current form.
Comments on the Quality of English LanguageThe English could be improved to more clearly express the research.
Author Response
"The English could be improved to more clearly express the research."
Dear reviewer, Thank you for the second round of revision. To increase the impact of our text, we performed a systematic review of the manuscript and restructured the discussion by checking the PRISMA statement for the manuscript.
We are attaching the new manuscript and highlighting the critical amendment sentences in green.
Our gratitude for the opportunity. Best regards